# Role of Extracellular Vesicles in Abdominal Aortic Aneurysm: Pathophysiology, Biomarkers, and Therapeutic Potentials

**DOI:** 10.3390/ijms27020567

**Published:** 2026-01-06

**Authors:** Kazuki Takahashi, Yusuke Yoshioka, Naoya Kuriyama, Shinsuke Kikuchi, Nobuyoshi Azuma, Takahiro Ochiya

**Affiliations:** 1Division of Molecular and Cellular Medicine, Institute of Medical Science, Tokyo Medical University, 6-7-1 Nishishinjuku, Shinjuku-ku, Tokyo 160-0023, Japan; k-takahashi11@asahikawa-med.ac.jp (K.T.); yyoshiok@tokyo-med.ac.jp (Y.Y.); 2Department of Vascular Surgery, Asahikawa Medical University, 1-1-1 Midorigaokahigashi2-jo, Hokkaido, Asahikawa-shi 078-8510, Japan; kuriyaman@asahikawa-med.ac.jp (N.K.); kikuchi@asahikawa-med.ac.jp (S.K.); nazuma@asahikawa-med.ac.jp (N.A.)

**Keywords:** abdominal aortic aneurysm, extracellular vesicles, biomarker, microRNA

## Abstract

Abdominal aortic aneurysm (AAA) is a life-threatening disease. Although AAA is generally asymptomatic, the mortality rate remains very high once rupture occurs, even with successful treatment. The pathophysiology of AAA involves inflammatory cell infiltration, smooth muscle cell apoptosis, and extracellular matrix degradation. However, there are various unclear aspects of pathophysiology due to cellular heterogeneity and multifactorial disease. Moreover, there are no blood biomarkers or available pharmacological drugs for AAA. Extracellular vesicles (EVs) are lipid bilayer particles released from every type of cell for intercellular communication. EVs include proteins, DNA, RNA (mRNA, microRNA), and lipids. EV cargos are delivered to recipient cells and modulate their biological effects. Although fewer studies have investigated EVs in AAA than in other cardiovascular diseases with similar molecular mechanisms, recent research indicates that EVs play a significant role in AAA development. Further research on EVs and AAA will contribute to the elucidation of AAA pathophysiology and the development of novel pharmacological drugs. In this review, we summarize the EV-associated pathophysiology, EV-based biomarkers, and EV-based treatment strategies in AAA. We also discuss the prospects for EVs research in AAA.

## 1. Introduction

Abdominal aortic aneurysm (AAA) is a life-threatening aortic disease with a risk of aortic rupture [1]. Including patients who fail hospitalization or those without treatment, the mortality rate for rupture is approximately 70% [2]. Epidemiological studies have shown that the prevalence in men over 65 years is 1–2% [3]. Risk factors of AAA are older age, male sex, smoking, hypertension, dyslipidemia, family history, and White race [1]. In contrast, the rupture rate of AAA is higher in women than in men [3]. AAA is generally asymptomatic, and diagnosis is based on imaging [1]. Abdominal ultrasonography (US) has high sensitivity, and US is the gold standard for screening [1]. However, due to low screening participation, the majority of AAA is detected incidentally during diagnostic CT and MRI for other diseases, such as cancer and cardiovascular disease (CVD) [2]. Therefore, a more convenient and low-cost diagnostic test is expected. The indications for treatment are a maximum AAA diameter > 50 mm in women and >55 mm in men, a symptomatic aneurysm, and rapid aneurysm growth [1]. The treatment options available are surgery or endovascular aneurysm repair (EVAR). However, both treatment options have mortality and complication risks. Although EVAR is minimally invasive, long-term outcomes remain under discussion [4]. Furthermore, surgical intervention for smaller AAAs has no life-extending effect, and management is limited to risk factor control, such as smoking cessation [5,6]. The primary pathogenesis of AAA involves inflammatory cell infiltration, vascular smooth muscle cell (VSMC) apoptosis, and extracellular matrix (ECM) degradation, and the molecular mechanisms underlying these processes have been reported [7]. Although the molecules involved in AAA development have been identified, the mechanism of AAA pathophysiology remains incompletely understood. Therefore, it is essential to elucidate these unresolved mechanisms of AAA through novel and comprehensive approaches.

Extracellular vesicles (EVs) are particles released from cells, delimited by a lipid bilayer, and incapable of replicating on their own [8]. The characteristics of released EVs reflect the physiological state of their cells of origin. Historically, EVs have been described using operational terms, including physical characteristics such as size (e.g., small/large EVs). However, size can overlap across EV subtypes and depends on the characterization method; therefore, EV subtype definitions should not rely on size alone. Until recently, terms such as exosomes and microvesicles were widely used [9]. However, the International Society for Extracellular Vesicles (ISEV) recommends using “EVs” unless the biogenesis pathway can be clearly demonstrated. Exosome is a biogenesis-related term indicating origin from the endosomal system. Ectosome is a biogenesis-related term indicating origin from the plasma membrane, and its size can overlap with that of exosomes; thus, size alone cannot distinguish these biogenesis-defined categories. EVs act locally, enter the bloodstream, and influence distant organs [10]. EVs contain multiple molecules, including proteins, messenger RNA (mRNA), microRNA (miRNA), DNA, lipids, and metabolites. Among these cargos, miRNAs are particularly important and have been widely studied [11,12]. The surface proteins and glycans of EVs are also dependent on the releasing cell and reflect cellular changes [13]. The lipid bilayer stabilizes EVs in the extracellular environment, protecting their cargos from endogenous DNases and RNases, thereby maintaining their stability. EVs have also been detected in virtually all bodily fluids and are being investigated as biomarkers [14]. For this reason, EVs and EV-associated miRNAs have been widely studied in cancer and many other research areas [15,16]. In the field of CVD, EVs are also associated with the regulation of vascular homeostasis and disease processes, and ongoing studies are exploring their pathophysiological roles and potential as novel pharmacological targets [17]. CVD includes coronary artery disease (CAD), aortic stenosis, cerebrovascular disease, and peripheral artery disease (PAD), many of which share major risk factors with AAA, such as hypertension and smoking, and frequently are comorbid in the same patients [18]. These CVDs also share several pathological features with AAA, including inflammatory cell infiltration and VSMC phenotypic switching [19]. The main difference is that AAA has progressive arterial dilatation. Despite these shared features, EV research specifically on AAA development remains limited compared with other CVD fields.

This narrative review summarizes current studies on EVs in AAA pathophysiology, biomarkers, and therapeutic approaches. In addition, based on evidence from other related research, we discussed the prospects for EVs in AAA research.

## 2. Pathophysiology of AAA, Cellular and Molecular Mechanisms

The aortic wall is composed of three layers—the intima, tunica media, and adventitia—and the principal cell types in each layer are endothelial cells (ECs), VSMCs, and aortic adventitial fibroblasts (AoAF), respectively. Among these layers, the tunica media is crucial for maintaining aortic integrity and consists of elastic fibers and VSMCs. It is also bounded by the internal and external elastic lamina [20]. In contrast, AAA is characterized by a dilated vascular diameter and destruction of elastic fibers. Another characteristic feature of AAA is the formation of an intraluminal thrombus (ILT), which exhibits a distinct structural morphology compared with aneurysms in other regions, such as thoracic aortic aneurysms (TAA) and cerebral aneurysms [21]. Several pathological processes contribute to AAA development, with major factors including inflammatory cell infiltration, VSMC apoptosis and phenotypic switching, and extracellular matrix (ECM) degradation [7]. These pathological changes involve a variety of cells and molecular mechanisms.

AAA tissue contains a variety of cell types, and single-cell analyses have identified multiple cellular populations involved in AAA pathogenesis. VSMC, EC, AoAF, and several inflammatory cells, including macrophages, T cells, B cells, neutrophils, and mast cells, are the major cell types present in AAA lesions [22,23]. Single-cell analysis also showed increased numbers of immune cells and decreased numbers of VSMCs and fibroblasts [22]. Cytokines, matrix metalloproteinases (MMPs), and other molecules produced by these cells also contribute to AAA formation [7]. Numerous miRNAs are also associated with these pathological mechanisms [23,24,25,26]. Below, we present an overview of the molecular mechanisms, with a particular focus on the cellular contributors (Figure 1).

Inflammatory cells are significantly associated with AAA development [23]. Among these, macrophage infiltration into the aortic wall is a hallmark pathological feature of AAA [7]. Traditionally, monocytes are thought to differentiate into two major macrophage subtypes—M1 and M2—both of which are upregulated in AAA tissues [23]. M1 macrophages produce chemokines and cytokines, such as interleukin-6 (IL-6), tumor necrosis factor-α (TNF-α), and monocyte chemoattractant protein-1 (MCP-1), as well as MMPs. M1 macrophages produce MMP9 and contribute to ECM degradation [7]. Furthermore, macrophages promote MMP activation in VSMCs [27]. The thioredoxin-interacting protein–NOD-like receptor protein 3 (TXNIP-NLRP3) inflammasome and damage-associated molecular patterns (DAMPs), such as High-Mobility Group Box 1 (HMGB1), are also associated with M1 macrophage activation [28,29]. In contrast, M2 macrophages secrete anti-inflammatory cytokines such as Transforming Growth Factor-β (TGF-β) and IL-10, contribute to ECM repair, and inhibit AAA progression. However, the M1/M2 ratio is higher in AAA tissue and is associated with AAA development [23].

Neutrophils are another significant source of MMPs in AAA [30]. Neutrophil elastase, released by activated neutrophils, exacerbates AAA by activating MMPs and inactivating their inhibitors [7]. Neutrophil extracellular traps (NETs) are released when neutrophils undergo a specific cell death program in response to stimuli such as IL-1β. NETs activate NLRP3 in macrophages, leading to increased production of IL-1β and IL-18 as well as further inflammatory responses in the aortic wall [31].

T cells, B cells, and mast cells have been reported to contribute to adventitial inflammation, which plays a role in AAA pathophysiology [32]. CD4^+^ T cells differentiate into Th1, Th2, Th17, and regulatory T (T reg) cells. Among these subsets, Th1 and Th17 cells are associated with AAA development [7]. Th1 cells secrete interferon-γ (IFN-γ), which promotes macrophage activation and recruitment, whereas Th17 cells enhance macrophage activity through IL-17 secretion [33]. In contrast, Th2 and Treg cells exert anti-inflammatory effects, but these cells decrease in AAA tissue. B cells also play an essential role in AAA by secreting antibodies. B cells mainly include B1, B2, and regulatory B cells [7]. B cell-derived immunoglobulins induce IL-6 and MMP9 secretion in the AAA lesion [34]. In B cell-depleted mice, the number of regulatory T cells increases, and AAA formation is suppressed [35]. Furthermore, B cell depletion suppresses macrophage activation [36]. Mast cells are associated with neovascularization in the aortic wall and macrophage activation [23,37]. Mast cells enhance the production of MMP9 by macrophages [38].

VSMCs are the principal component of the aorta and play a role in maintaining aortic structure to synthesize ECM [7]. ECM degradation is a major pathological feature of AAA. MMPs and a disintegrin and metalloproteinase (ADAM) family contribute to ECM degradation [39]. In particular, the upregulation of MMP2 and MMP9 promotes AAA formation [40]. On the other hand, VSMCs preserve vascular homeostasis by releasing tissue inhibitors of metalloproteinases (TIMPs), which modulate MMP activity [41]. Phenotypic switching of VSMCs is another crucial factor in AAA development [42]. Under normal physiological conditions, VSMCs exhibit a contractile phenotype. However, under injury or inflammatory conditions, VSMCs switch to a synthetic phenotype [7]. Several miRNAs have also been implicated in VSMC phenotypic switching, such as miR-15a and miR-3154 [24,43]. Synthetic VSMCs secrete inflammatory mediators and MMPs, whereas TGF-β promotes the maintenance of the contractile phenotype [44]. Decreasing VSMC is also associated with AAA formation. Multiple factors promote VSMC apoptosis, including IL-6, TNF-α, and MCP-1. In addition, pyroptosis and ferroptosis of VSMCs also contribute to AAA formation [7].

ECs constitute the inner layer of the aorta and play a crucial role in maintaining aortic homeostasis. Stimulated ECs release reactive oxygen species (ROS), which contribute to AAA formation [7]. ROS promotes ECM remodeling and induces VSMC apoptosis in AAA [45]. ECs secrete chemokines such as C-X-C motif chemokine ligand 1 (CXCL1) and CXCL8, which recruit inflammatory cells to the aortic wall [30]. Cigarette smoke-induced endothelial dysfunction further enhances monocyte recruitment, activates inflammatory signaling pathways, and amplifies vascular injury [46]. In addition, ECs are also constantly exposed to biomechanical shear stress from blood flow [7]. Under conditions of excessive shear stress, ECs exhibit increased MMP activity and reduced TIMP activity [47], further contributing to ECM degradation and AAA development.

AoAF is most abundant cell in the vascular adventitia [48]. However, compared with other vascular cells, the studies on AoAF in AAA have been limited. Suppressing adventitial fibroblast proliferation and collagen synthesis increased the risk of AAA rupture [49]. Furthermore, AoAF stimulated by angiotensin II releases cytokines that promote monocyte infiltration into the aortic wall [50]. Reduced ECM production has been observed in AoAF in thoracic aortic aneurysm (TAA) tissue [51]. In both human AAA tissue and mouse AAA tissue, AoAF undergoes phenotypic transformation into myofibroblasts [52]. Myofibroblasts are also implicated in aneurysm formation in TAA [53].

This section reviewed the cellular and molecular mechanisms associated with AAA development. Many cell types and molecular mechanisms are involved, primarily centered on inflammation and ECM degradation. Additional cell populations not detailed here may also play important roles. However, although individual pathological mechanisms have been described, a coherent sequence linking risk factors such as smoking exposure to cellular alterations, intercellular interactions, and ultimately to AAA formation has not yet been fully defined. Furthermore, the absence of a clear pathophysiology has hindered the development of effective pharmacological therapies. As mediators of intercellular communication, including various molecular cargoes, EVs may provide new insights into these unresolved mechanisms and support future therapeutic development. Therefore, in the following sections, we first summarize the current evidence on EVs and AAA to help address these unresolved problems.

## 3. EV-Associated AAA Pathophysiology

The pathophysiological roles of EVs have been investigated in various disease fields [15,16,17]. Some studies have reported an association between EVs and AAA. We reviewed not only existing research on AAA and EV but also EV-related pathogenesis research that may be related to AAA development (Figure 2).

Several animal studies have demonstrated that EVs contribute to AAA development in the AAA mouse model [54,55,56]. EVs released by inflammatory cells are implicated in AAA development. Macrophage-derived EVs enhance AAA development, and VSMCs exposed to these EVs show increased MMP2 expression through activation of the JNK and p38 signaling pathways [54]. Furthermore, M1 macrophage-derived EVs promote inflammation and pyroptosis in VSMCs. M1 macrophage-derived EVs contain the LncRNA PVT1, which suppresses miR-186-5p and regulates HMGB1 expression, leading to promotion of VSMC inflammation and pyroptosis [55]. T cell-derived EVs promote macrophage recruitment to the aorta. These EVs contribute to AAA pathogenesis and macrophage migration into the aortic wall. Pyruvate kinase muscle isozyme 2 (PKM2)-activated T cell-derived EVs induce iron-dependent lipid peroxidation in macrophages, leading to promoting their migration into the aortic wall. In addition, plasma EVs of AAA patients induced macrophage infiltration into the aortic wall [56].

Flow cytometric analysis of EVs in AAA tissue conditioned medium revealed distinct EV populations. ILT and AAA wall-derived EVs contained higher levels of CD41^+^ platelet-derived EVs and CD14^+^ monocyte-derived EVs compared with the healthy aortic wall. Furthermore, CD41^+^ platelet-derived EVs contained elevated levels of ficolin-3 [57]. The detection of these EVs not only in the ILT but also within the aneurysmal wall suggests that platelet- and monocyte-derived EVs may contribute to AAA development processes. Neutrophil-derived EVs also participate in AAA pathogenesis. ADAM10- and ADAM17-positive EVs are released from ILTs, with neutrophil-derived EVs predominantly expressing these molecules, whereas platelet-derived EVs do not. Exposure of a human neutrophil to tobacco smoke extract induces the release of ADAM10- and ADAM17-positive EVs [58]. This finding suggests that EVs may represent a link between smoking and AAA pathophysiology. miRNAs present in blood and AAA tissue-derived EVs have also been implicated in AAA-related pathology. miR-106a is elevated in both blood EVs and AAA tissue EVs. Although the cell type of origin for EVs has not been identified, miR-106a decreased VSMC viability, increased apoptosis, reduced TIMP2 expression, and increased MMP2 and MMP12 secretion in VSMCs [59].

At present, only immune cell-derived EVs have been reported to be associated with AAA. However, findings from other research fields suggest that EVs may participate in various pathophysiological mechanisms of AAA development [17]. Stimulated ECs-derived EVs promote VSMC proliferation and phenotypic switching in atherosclerosis [60,61]. ECs also release pro-inflammatory mediators, including chemokines and cytokines such as Intercellular Adhesion Molecule 1 (ICAM-1), C-C motif chemokine ligand 2 (CCL2), and CXCL10. The ICAM-1^+^ EV subpopulation promotes THP-1 monocyte migration [62]. Fibroblast-derived EVs inhibit VSMC proliferation via miR-155 [63]. Cigarette smoke-induced cell-derived EVs exhibit altered proteomic profiles [64]. Moreover, smoke-exposed monocytes and macrophage-derived EVs promote inflammatory and proteolytic processes [65]. In TAA and aortic dissection, mechanical stretching induces EV release from VSMCs, leading to endoplasmic reticulum stress and endothelial cell dysfunction [66]. These findings suggest that EV-mediated mechanisms observed in other diseases may also be relevant to AAA.

Collectively, these studies indicate that EVs may play essential roles in the development of aortic aneurysms. In particular, several studies have shown that certain EVs can activate macrophages. In addition, macrophage-derived EVs promote key pathological processes, including extracellular matrix degradation and VSMC death. However, EVs derived from other cell types implicated in AAA development, such as ECs and AoAF, remain poorly understood. Therefore, further investigations are required to elucidate the molecular mechanisms linking EVs to AAA pathophysiology.

## 4. EV-Biomarkers of Diagnosis, and Follow-Up of AAA

There are currently no specific blood biomarkers for AAA in real-world clinical practice, and US and CT remain the gold standards for diagnosing and monitoring AAA. Numerous studies have explored potential blood biomarkers, including standard clinical blood tests [67], proteomics-based approaches [68], genomic analyses [69], and circulating miRNAs [70]. EV-associated ficolin-3, which is implicated in AAA pathogenesis, has also been proposed as a potential plasma biomarker [57].

Furthermore, AAA requires a wide range of biomarkers, not only for AAA diagnosis but also for post-treatment evaluation [71] and for predicting rupture risk [72]. However, none of them have yet been implemented in clinical practice. EV-based biomarker research has been conducted in various fields [73], and EV-derived biomarkers for AAA are currently under investigation (Table 1).

Proteins contained in EVs were examined as potential diagnostic biomarkers for AAA [74,75]. Proteomics analysis revealed that oncoprotein-induced transcript 3 (OIT3), dermcidin, annexin A2, platelet factor 4 (PF4), ferritin, and C-reactive protein (CRP) were elevated in plasma EVs from AAA patients [74]. In addition, proteomic analyses have identified other EV-associated biomarkers, showing that IL-4, IL-6, and oncostatin M levels were elevated, whereas neurturin and MCP-1 levels were reduced in EVs from AAA patients [75]. miRNAs in EVs have also been investigated as biomarkers for AAA [59,76,77]. Several miRNAs in plasma EVs were found to be either upregulated or downregulated. miR-106, miR-29, and miR-33 were upregulated, whereas miR-204 and miR-24 were significantly downregulated compared with healthy donors [59]. Lopez et al. identified 40 differentially expressed miRNAs in plasma EVs from AAA patients. miR-122-5p was the most strongly altered and significantly downregulated in AAA [76]. Additionally, given that AAA shares many risk factors with coronary artery disease (CAD) [18], comparing EV-derived biomarkers across different atherosclerotic diseases is essential. Hildebrandt et al. demonstrated that EV-derived miRNAs are specific to AAA when compared with other CVDs. In their analysis, 27 miRNAs were significantly different between CVD patients and healthy controls. The CVD cohort included individuals with CAD, carotid stenosis, AAA, and PAD, and each condition exhibited its own distinct pattern of EV miRNAs relative to controls. Among these, miR-122-5p, miR-2110, and miR-483-5p were specifically upregulated in patients with only AAA [77].

EV-based biomarkers have also been examined for evaluating complications such as endoleak (EL) after EVAR and postoperative paraplegia [78,79]. Patients who developed EL showed an increase in activated CD31^+^ endothelial-derived EVs at 1 month after EVAR and a decrease in activated CD41^+^ platelet-derived EVs at 6 months post-procedure in flow cytometry EVs analysis [78]. This research indicated the possibility of non-radiation-based follow-up methods after EVAR. In addition, postoperative increases in neuronal-derived blood EVs may predict lower extremity weakness (LEW) and paraplegia after branched EVAR (BEVAR). This phenomenon has been suggested to be associated with postoperative acute insulin resistance [79].

This section summarizes EV biomarkers in AAA. However, EV biomarker research remains limited, and most available studies include only a small number of cases. Furthermore, no studies have yet examined EV cargos to identify biomarkers that can distinguish small and large AAAs or predict rupture risks. Nevertheless, the development of blood-based EV biomarkers offers a more convenient screening approach than US or CT. Because blood tests can be implemented in a high-throughput manner, EV-based assays may facilitate population-level screening and improve accessibility. In addition, EV-based biomarkers may offer higher diagnostic accuracy than conventional blood tests and hold potential clinical value for AAA screening and rupture risk assessment. On the other hand, several studies of circulating miRNAs that do not specifically account for EVs have attempted to identify such biomarkers [70]. Validating these circulating miRNAs and other components within EVs may provide new insights and discover more accurate, clinically relevant biomarkers.

## 5. Future EV-Based Treatment for AAA

Many pharmacological drugs have been shown to suppress AAA in animal studies. However, antihypertensive, antibiotic, antiplatelet, and anti-inflammatory medications that demonstrated inhibitory effects on AAA formation in mouse models have failed to show clinical efficacy in randomized trials (RCTs) [22]. Recently, statins and the antidiabetic drug metformin have attracted attention as potential pharmacological therapies for AAA. Systematic reviews and meta-analyses suggest that these agents may be associated with slower AAA progression [80]. Real-world pharmacovigilance data-based Mendelian randomization studies suggest that statin use may suppress AAA development [81]. However, no RCTs have definitively demonstrated the efficacy of statins in inhibiting AAA development. In contrast, RCTs of metformin are currently underway, and their results are awaited [82]. Thus, there remains a significant unmet need for pharmacological drugs that can inhibit aneurysm enlargement and prevent rupture. Novel modalities, including nucleic acid drugs [83], cell therapy [84], and nanoparticles [85], have recently attracted attention and have been shown to suppress AAA development in animal studies. Nevertheless, each of these approaches presents important limitations. Although studies are underway to deliver miRNAs directly [25], free miRNAs are rapidly degraded and exhibit poor cellular uptake [14]. Mesenchymal stromal cells (MSCs) have also been reported to have therapeutic effects in various fields, including AAA [86]. However, the direct administration of MSCs has been shown to involve risks of cancer cellularization [87] and lung embolization [88]. In contrast, EV and EV-associated miRNAs can overcome these limitations [14]. Furthermore, MSC-EV administration is as effective as MSC administration in an AAA mouse model [84]. Given their high stability and effectiveness, EVs have attracted considerable attention as therapeutic tools in AAA research.

To date, EVs from bone marrow-derived MSCs (BM-MSCs), adipose-derived MSCs (AD-MSCs), and iPSC-derived MSCs (iPS-MSCs) have been investigated for their therapeutic potential in AAA. Although the underlying genes, proteins, and signaling pathways differ among studies, these MSC-derived EVs consistently suppress AAA development in animal models (Table 2).

**Table 2 ijms-27-00567-t002:** EV-based therapeutic approaches for AAA in animal models.

Cell Source	EV Isolation Method	Animal Model	EV Dose (μg, Particles)	EV Administration Routes	Therapeutic Effect in AAA	Reference
BM-MSCs	Ultracentrifugation	Angiotensin II mouse models	10 μg, Single dose after 4 weeks of Ang II stimulation	Injected into the tail veins	IL-1β, TNF-α, MCP-1 decreased MMP2/MMP9 activity downregulated TIMP2 and IGF-1 increased M1 macrophages decreased M2 macrophages increased	[89]
MSCs (origin not specified)	Ultracentrifugation	Angiotensin II mouse models, CaCl_2_ mouse model	3 μg/g, once a week for 28 days	Injected into the tail veins	M1 macrophages decreased M2 macrophages increased	[90]
AD-MSCs	Ultracentrifugation	Angiotensin II mouse models	100 μg, every 3 days for 28 days	Injected into the tail veins	IL-1β, IL-18 decreased TXNIP-positive macrophages decreased	[91]
MSCs (origin not specified)	Ultracentrifugation	Elastase mouse model	54 μg, Single dose 1 day after elastase treatment	Injected into the tail veins	IL-17, IL-23, IFN-γ, TNF-α, RANTES, MCP-1, KC, MIP-1α decreased HMGB-1 decreased macrophage, neutrophil, and CD3^+^ T cell infiltration decreased	[92]
BM-MSCs	Ultracentrifugation	Elastase rat models	No	No	Unknown	[93]
AD-MSCs	Ultracentrifugation	Angiotensin II mouse models	5 × 10^9^ particles, every 3 days for 28 days	Injected into the tail veins	IL-6, CCL2 decreased MMP2/MMP9 activity downregulated ROS level reduced M1 macrophages decreased Senescence proteins (p16, p21) decreased	[94]
iPS-MSCs	Anion exchange chromatography	Angiotensin II mouse models	2 × 10^10^ particles, once a week for 28 days	Injected into the tail veins	inhibit VSMC senescence	[95]
BM-MSCs	Ultracentrifugation	Angiotensin II mouse models	7 × 10^10^ particles, Single dose 1 day after Ang II stimulation	Injected into the tail veins	Inhibit NETosis Inhibit VSMC ferroptosis	[96]
BM-M2 macrophage	Ultracentrifugation	Angiotensin II mouse models	100 μg, 3 times per week for 28 days	Injected into the tail veins	TNF-α, IL-1β, iNOS, ICAM1 decreased M1 macrophage decreased	[97]

In vitro studies have elucidated the modes of action of MSC-derived EVs (Figure 3). MSC-derived EVs have been shown to suppress AAA development by inhibiting aortic inflammation and macrophage activation in AAA mouse models. In mice treated with MSC-derived EVs, expression of proinflammatory cytokines such as IL-1β, TNF-α, and MCP-1 was decreased in AAA tissue, and ROS levels were also reduced. Furthermore, MMP2 and MMP9 activities were downregulated, whereas TIMP2 and IGF-1 were increased compared with controls. The MSC-EV-treated group also showed reduced percentages of M1 macrophages and increased percentages of M2 macrophages, indicating M2 polarization [89,90,91,92,94].

MSC-derived EV primarily suppresses M1 macrophage activity. BM-MSC-derived EV reduced the expression of MMP-2, MMP-9, TNF-α, and IL-1β in M1 macrophage [89]. MSC-derived EVs also suppress CD74 expression and modulate the TSC2–mTOR–AKT signaling cascade in M1 macrophages [90]. In addition, MSC cargo also contributes to macrophage regulation. BM-MSC-derived EVs included AAA-suppressive miRNAs, such as miR-24 [26]. AD-MSC-derived EVs also inactivate TXNIP–NLRP3 inflammasome via miR-17-5p and inhibit pyroptosis in M1 macrophages [91]. Furthermore, miR-147 also attenuated HMGB1 secretion in M1 macrophages [92].

VSMC inflammation is also a therapeutic target of MSC-derived EVs. MSC-derived EVs reduced cytokine production, decreased MMP2 production, and increased TIMP2 in VSMC [89,92,93]. AD-MSCs inhibited VSMC senescence and suppressed AAA development. The effect was modulated by miR-19b-3p, which suppresses the MST4/ERK/Drp1 pathway in VSMCs [94]. iPS-MSC-EVs similarly suppress VSMC senescence. Furthermore, to enhance the therapeutic efficacy of MSC-derived EVs, Ouyang et al. explored the use of genetically modified MSC-EVs. Nicotinamide phosphoribosyl transferase (NAMPT), which is significantly downregulated in AAA, was overexpressed in iPS-MSC-derived EVs using a lentivirus vector. These NAMPT-enriched iPS-MSC-derived EVs more effectively inhibited VSMC senescence and suppressed AAA development compared with non-modified MSC-EVs [95].

MSC-derived EVs also influence other inflammatory cell populations in the AAA model mouse. MSC-derived EVs decrease IL-17 expression in CD4^+^ T cells [92]. In addition, BM-MSC-derived EVs suppress NET formation in neutrophils and inhibit VSMC ferroptosis [96].

In addition to MSC-derived EVs, EVs from M2 macrophages differentiated from mouse bone marrow (BM)-derived monocytes have also been reported to suppress AAA development. M2 macrophage-derived EVs inhibited M1 polarization through the PARP-1/PP-1α/JNK/c-Jun signaling pathway via miR-221-5p [97].

Nakazaki et al. suggested the potential use of EVs for treating complications associated with AAA therapy. MSC-derived EVs may be effective in treating paraplegia caused by spinal cord ischemia, a severe complication that can occur after AAA or thoracoabdominal aortic aneurysm surgery. MSC-derived EVs have been shown to promote functional recovery in experimental models of traumatic spinal cord injury [98]. Although ischemia and trauma involve different pathological mechanisms, these findings suggest that MSC-derived EVs have potential as a therapeutic strategy for paraplegia after aneurysm treatment.

Although the therapeutic potential of EVs for AAA has been demonstrated, several challenges remain for EV-based drug development, including issues related to administration routes, dosing, and off-target effects [99,100,101]. Intravenous administration is currently considered the most common method. In mice, systemically administered EVs have a short blood half-life of approximately 2 min, whereas in primates, the circulation time is approximately 60 min [99,100]. Following intravenous administration, EVs are rapidly taken up by organs and are not readily degraded in the circulation. At low doses, EVs are preferentially accumulated in organs such as the liver and spleen. However, higher doses lead to broader distribution and accumulation in additional organs [101]. This dose-dependent biodistribution presents a major challenge for therapeutic applications, as effective delivery of EVs to the aorta may require the administration of large quantities of EVs in humans. Therefore, several studies have focused on enhancing the specificity and efficiency of EV uptake by target tissues.

One approach is to use EVs derived from cells that share the same origin as the target tissue or cell type. MSC-derived EVs are equally taken up by heart, lung, and skin fibroblasts. In contrast, cardiomyocyte-derived EVs are preferentially internalized by cardiac fibroblasts compared with lung or skin fibroblasts [102]. Another approach is to enhance cellular uptake by modifying EVs. The expression of a cardiac-targeting peptide (CTP) fused to Lamp2b on the EV membrane can improve EV delivery to cardiac cells [103]. In addition, MSC-derived EVs modified with a short synthetic peptide were more efficiently taken up by VSMCs isolated from AAA model rats [104]. These strategies may help overcome current limitations in EV-based drugs.

Specific delivery strategies are also being explored in the CVD field to target vascular sites with EVs or EV-related therapeutics. These approaches include the use of stents or stent grafts. Schneider et al. first suggested the effectiveness of endovascular seeding of BM-MSCs for the treatment of AAA [105]. Subsequently, anti-miRNA drugs were delivered to the target vessel using expandable balloons and drug-eluting stents [106], and a nanocarrier was also delivered via a stent graft [107]. Furthermore, in a mouse model of arterial disease, EV-coated stents accelerated reendothelialization and reduced in-stent restenosis [108], indicating progress in EV-delivery technologies for stent delivery systems. Taken together, these findings suggest that vascular disease-specific delivery approaches already exist and that their application in EV-based therapy holds substantial therapeutic potential.

Beyond biodistribution and targeting, the physicochemical stability of EV products during storage and handling (e.g., temperature control, freeze–thaw cycles, and potential aggregation or loss of bioactivity) is also an important consideration for clinical translation. Sterile aqueous suspensions are commonly used for intravenous injection/infusion in preclinical studies, whereas lyophilized formulations have been explored as a potential approach to improve shelf-life and facilitate handling [109]. While these formulation topics remain an active area of development, they represent practical constraints that must be addressed alongside dosing and targeting to enable EV-based therapeutics for AAA. In addition, scalable GMP-compatible manufacturing and robust quality control (e.g., identity, purity, potency, sterility, and batch-to-batch consistency) remain major hurdles for clinical translation of EV-based therapeutics [110].

EV-based drugs are studied in various fields, and multiple clinical trials are currently underway worldwide. According to the U.S. National Institutes of Health ClinicalTrials database, https://clinicaltrials.gov (accessed on 20 November 2025), 177 clinical trials involving EVs as therapeutic agents were registered. However, several challenges must still be resolved, including the identification of optimal cell sources, standardization of EV isolation methods, and clarification of EV–cell interaction mechanisms [111]. Although no EV-based drugs are currently available, EVs have attracted considerable attention as potential therapeutic drugs. At present, there have been no clinical trials of EV therapy for AAA. Nevertheless, EV-based treatment is expected to emerge as a promising therapeutic strategy for AAA, and further research will be essential to advance this field.

## 6. Conclusions

EVs are now recognized as important mediators of intercellular communication and have been implicated in the pathogenesis of AAA. This review summarizes the current understanding of the roles of EVs in AAA development and highlights their potential as biomarkers for disease diagnosis and monitoring, and therapeutic agents.

Consistent with the scope of this review focusing on EV biology in AAA, studies in pathophysiology investigating EVs in AAA remain limited, with most reports focusing primarily on EVs derived from inflammatory cells and VSMC. However, AAA pathophysiology involves multiple cell types and complex molecular mechanisms; therefore, further studies on EVs derived from other relevant cell types are warranted. These cell-derived EVs are needed for further research. In addition, the functional roles of EV cargo remain incompletely understood, and mechanistic studies are needed to clarify how EVs contribute to AAA initiation and progression.

EVs also have potential as biomarkers for AAA diagnosis. Although EV-based biomarkers show promise, further research is required to support their clinical application. Previous studies have been limited by small sample sizes, and large-scale human studies are needed. Moreover, the performance of EV-based biomarkers must be validated across multiple biological specimens before they can be translated into clinical practice. Despite these challenges, EV-based biomarkers have the potential to transform current strategies for AAA diagnosis, screening tests, and risk assessment.

In therapeutic applications, EVs may contribute to suppressing AAA development. However, current evidence remains limited, and several challenges must be addressed, including standardizing EV sources, dosing, and administration protocols. Moreover, the precise mechanisms underlying EV-mediated suppression of AAA remain incompletely elucidated. In addition, the large amounts of EVs potentially required for human administration, together with the associated medical costs, represent significant barriers to clinical translation. Nevertheless, no disease-modifying pharmacological therapy has been definitively established to halt AAA progression, and EVs may represent a promising avenue for future therapeutic development.

Although research on EVs in AAA remains limited compared with other fields, their clinical potential is considerable. Further studies are required to advance their translation into future clinical applications.

## Figures and Tables

**Figure 1 ijms-27-00567-f001:**
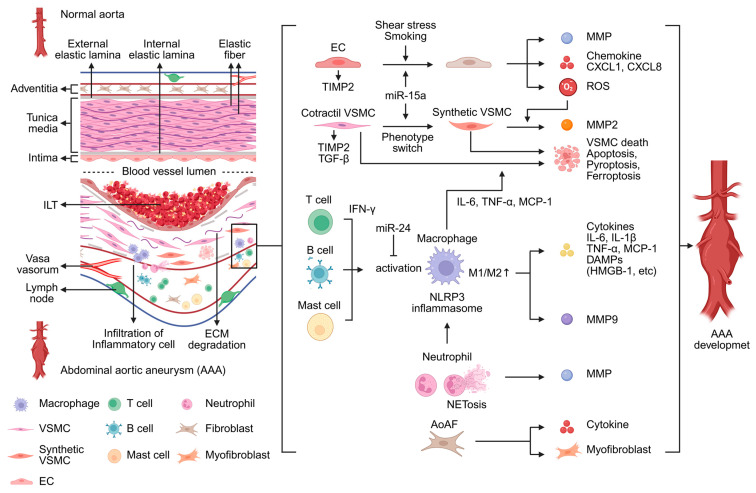
The normal aorta exhibits a three-layered structure with well-preserved elastic fibers. In contrast, abdominal aortic aneurysm (AAA) is characterized by extracellular matrix (ECM) degradation and infiltration of inflammatory cells, accompanied by the formation of an intraluminal thrombus (ILT). AAA development is driven by multiple pathological processes, including inflammatory cell infiltration, vascular smooth muscle cell (VSMC) apoptosis and phenotypic switching, as well as ECM degradation. These pathological changes involve complex interactions among diverse cell types and molecular mechanisms. The figure was created in BioRender. Takahashi, K. (2025) https://BioRender.com/y75hwzj (accessed on 24 December 2025). CXCL, C-X-C motif chemokine ligand; DAMPs, damage-associated molecular patterns; EC, endothelial cell; AoAF, aortic adventitial fibroblasts; HMGB-1, high-mobility group box 1; IFN-γ, interferon-γ; IL, interleukin; MCP-1, monocyte chemoattractant protein-1; MMP, matrix metalloproteinases; NET, neutrophil extracellular trap; NLRP3, NOD-like receptor protein 3; ROS, reactive oxygen species; TIMP2, tissue inhibitor of metalloproteinase 2; TNF-α, tumor necrosis factor-α.

**Figure 2 ijms-27-00567-f002:**
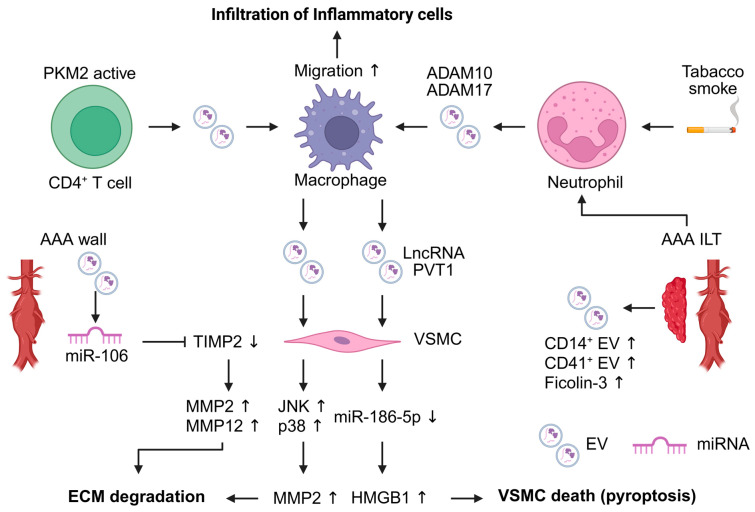
Extracellular vesicles (EVs) contribute to the pathogenesis of abdominal aortic aneurysm (AAA). EVs are mainly released from inflammatory cells within AAA lesions and act as mediators of intercellular communication. Macrophage-derived EVs influence vascular smooth muscle cells (VSMCs), leading to extracellular matrix (ECM) degradation and VSMC pyroptosis. EVs derived from T cells and neutrophils enhance macrophage migration. EV-associated microRNAs are involved in the upregulation of matrix metalloproteinases (MMPs) in VSMCs. The figure was created in BioRender. Takahashi, K. (2025) https://BioRender.com/a6o1rrz (accessed on 24 December 2025). ADAM, a disintegrin and metalloproteinase domain-containing protein; HMGB1, high mobility group box 1; ILT, intraluminal thrombus; JNK, c-Jun N-terminal kinase; PKM2, pyruvate kinase muscle isozyme 2; TIMP2, tissue inhibitor of metalloproteinase 2.

**Figure 3 ijms-27-00567-f003:**
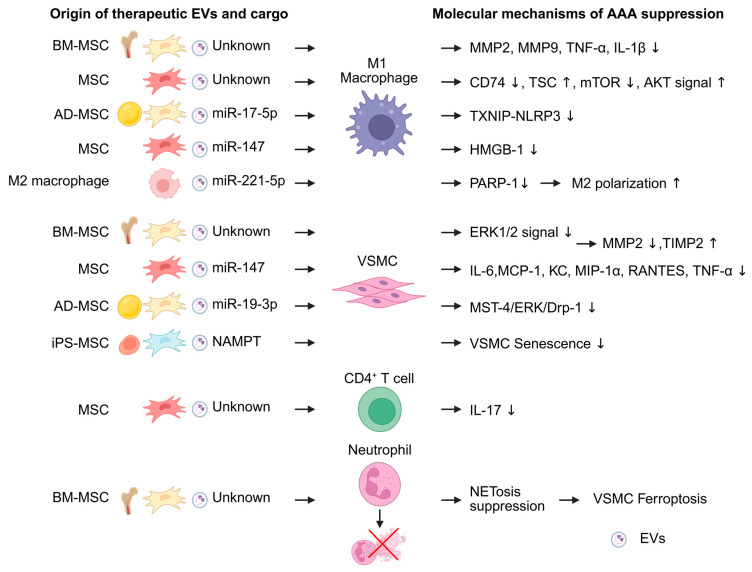
Extracellular vesicles (EVs) exert attenuating effects on abdominal aortic aneurysm (AAA) formation. Among them, EVs derived from mesenchymal stromal cells (MSCs) are the most extensively reported. MSC-derived EVs act on M1 macrophages, vascular smooth muscle cells (VSMCs), and neutrophils to suppress AAA development. In addition, EVs derived from M2 macrophages influence M1 macrophage polarization. MicroRNAs are mainly reported as the functional cargoes of these EVs. The figure was created in BioRender. Takahashi, K. (2025) https://BioRender.com/0tikhev (accessed on 24 December 2025). AD-MSC, adipose-derived MSC; BM-MSC, bone marrow-derived MSC; iPSMSC, iPS-derived MSC; HMGB1, high-mobility group box 1; KC, keratinocyte chemoattractant; MCP-1, monocyte chemoattractant protein-1; MST4, mammalian sterile-20-like kinase 4; NAMPT, nicotinamide phosphoribosyl transferase; NET, Neutrophil extracellular trap; mTOR, mechanistic target of rapamycin; RANTES, regulated upon activation, normal T cell expressed and secreted; TXNIP-NLRP3, thioredoxin-interacting protein–NOD-like receptor protein 3.

**Table 1 ijms-27-00567-t001:** EV-related AAA biomarkers.

Body Fluids	Number of Non-AAA/AAA (*n* = Number)	EV Isolation Method	Detection Method	Clinical Evaluation	Potential Biomarkers	Reference
Plasma	10/10	Differential centrifugation	Proteomics analysis	AAA diagnosis	OIT3, dermcidin, AnnexinA2, PLF4, Ferritin, CRP up-regulation	[74]
Plasma	9/18	Polymer-based precipitation method	Proteomics analysis and ELISA	AAA diagnosis	IL-4, IL-6, and oncostatin M up-regulation neurturin, MCP-1 down-regulation	[75]
Plasma	8/21	Ultracentrifugation	qPCR	AAA diagnosis	miR-106, miR-29, miR-33 up-regulation miR-204, miR-24 down-regulation	[59]
Plasma	11/15	Density gradient ultracentrifugation	NGS and qPCR	AAA diagnosis	miR-122-5p down-regulation	[76]
Serum	28/35	Column-based isolation	NGS	AAA diagnosis	miR-122-5p, miR-2110, miR-483-5p up-regulation	[77]
Plasma	After EVAR AAA *n* = 22	Unknown	Flow cytometric analysis	EVAR follow-up	EL: 1M endothelial-derived EV up-regulation 6M platelet-derived EVs down-regulation	[78]
Plasma	After BEVAR AAA *n* = 10	Polymer-based precipitation method	ELISA	Complication after BEVAR	LEW: neuron-derived blood EVs down-regulation	[79]

## Data Availability

No new data were created or analyzed in this study. Data sharing is not applicable to this article.

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
