# Peer review of "Role of Extracellular Vesicles in Abdominal Aortic Aneurysm: Pathophysiology, Biomarkers, and Therapeutic Potentials"

_ijms, 2026, doi:10.3390/ijms27020567_

Round 1
Reviewer 1 Report
Comments and Suggestions for Authors
Dear Authors,
I have made some comments in the txt of the manuscript, that you will find in attachment here. There were several overstatements that I highlighted. Some details to add, like the size of EV and adding statins as potential drug treatment for AAA. And improving the Conclusion. Regards

I have highlighted with comment some english wording that can be improved.
Author Response
Author's Reply to the Review Report (Reviewer 1)
I have made some comments in the txt of the manuscript, that you will find in attachment here. There were several overstatements that I highlighted.
We appreciate the reviewer for the careful review of our manuscript and for highlighting several overstatements in the text. We have revised the manuscript accordingly, and the suggested points have been addressed.
Line 23
The original sentence:
Although studies focusing on EVs in AAA are still limited compared with those in cardiovascular disease, which shares similar molecular mechanisms, EVs have significantly contributed to AAA development.
has been revised to:
Although fewer studies have investigated EVs in AAA than in other cardiovascular diseases with similar molecular mechanisms, recent research indicates that EVs play a significant role in AAA development.
Line 33
The original sentence:
Abdominal aortic aneurysm (AAA) is a life-threatening aortic disease that causes fatal due to aortic rupture
has been revised to:
Abdominal aortic aneurysm (AAA) is a life-threatening aortic disease with a risk of aortic rupture.
Line 55
problems → mechanisms
Line 79
use regarding novel pharmacological drugs → potential as novel pharmacological targets
Line 197: This subject → This section
Line 207 As suggested by the reviewer, we have added an appropriate reference to support this statement in the revised manuscript.
AAA mouse model [54-56].
Line 214
The original sentence:
EVs released by inflammatory cells promote AAA development.
has been revised to:
EVs released by inflammatory cells are implicated in AAA development.
Line 222
The original sentence:
These EVs promote AAA development and macrophage migration into the aortic wall.
has been revised to:
These EVs contribute to AAA pathogenesis and macrophage migration into the aortic wall.
Line 291 Proteomics analysis revealed that oncoprotein-induced transcript 3 (OIT3)~
We appreciate the reviewer’s suggestion to mention pathways that are up- or down-regulated. However, in the cited reference, the authors primarily reported changes in the expression of specific proteins and did not comprehensively analyze or indicate the associated signaling pathways. We therefore limited the description to the reported protein-level changes.
Some details to add, like the size of EV, and adding statins as potential drug treatment for AAA.
We appreciate the reviewer’s comment regarding EV size. We agree that size can be a useful descriptive parameter; however, MISEV2023 emphasizes that EV subtype definitions should not be based on size alone, and that measured size can vary depending on the characterization method. Accordingly, we revised the Introduction to clarify that size overlaps across EV subtypes and should not be used as the sole basis for classification.
In addition, in the AAA literature, EV size is not consistently reported with comparable methods across studies. Therefore, we did not add a separate size-based classification, and we used the terminology reported in the original studies (e.g., “EVs” or “small EVs” where appropriate).
Line 57-
Extracellular vesicles (EVs) are particles released from cells, delimited by a lipid bilayer, and incapable of replicating on their own [8]. The characteristics of released EVs reflect the physiological state of their cells of origin. Historically, EVs have been described using operational terms, including physical characteristics such as size (e.g., small/large EVs). However, size can overlap across EV subtypes and depends on the characterization method; therefore, EV subtype definitions should not rely on size alone. Until recently, terms such as exosomes and microvesicles were widely used [9]. However, the International Society for Extracellular Vesicles (ISEV) recommends using “EVs” unless the biogenesis pathway can be clearly demonstrated. Exosome is a biogenesis-related term indicating origin from the endosomal system. Ectosome is a biogenesis-related term indicating origin from the plasma membrane, and its size can overlap with that of exosomes; thus, size alone cannot distinguish these biogenesis-defined categories.
In terms of statins, we have described their potential role as a pharmacological treatment for AAA in the revised manuscript.
Line337
Many pharmacological drugs have been shown to suppress AAA in animal studies. However, antihypertensive, antibiotic, antiplatelet, and anti-inflammatory medications that demonstrated inhibitory effects on AAA formation in mouse models have failed to show clinical efficacy in randomized trials (RCTs) [22]. Recently, statins and the antidiabetic drug metformin have attracted attention as potential pharmacological therapies for AAA. Systematic reviews and meta-analyses suggest that these agents may be associated with slower AAA progression [80]. Real-world pharmacovigilance data–based Mendelian randomization studies suggest that statin use may suppress AAA development [81]. However, no RCTs have definitively demonstrated the efficacy of statins in inhibiting AAA development. In contrast, RCTs of metformin are currently underway, and their results are awaited [82].
As suggested by the reviewer, the Conclusion has been revised and expanded to improve clarity and readability.
EVs are now recognized as important mediators of intercellular communication and have been implicated in the pathogenesis of AAA. This review summarizes the current understanding of the roles of EVs in AAA development and highlights their potential as biomarkers for disease diagnosis and monitoring, and therapeutic agents.
Studies in pathophysiology investigating EVs in AAA remain limited, with most reports focusing primarily on EVs derived from inflammatory cells and VSMC. However, AAA pathophysiology involves multiple cell types and complex molecular mechanisms; therefore, further studies on EVs derived from other relevant cell types are warranted. These cell-derived EVs are needed for further research. In addition, the functional roles of EV cargo remain incompletely understood, and mechanistic studies are needed to clarify how EVs contribute to AAA initiation and progression.
EVs also have potential as biomarkers for AAA diagnosis. Although EV-based biomarkers show promise, further research is required to support their clinical application. Previous studies have been limited by small sample sizes, and large-scale human studies are needed. Moreover, the performance of EV-based biomarkers must be validated across multiple biological specimens before they can be translated into clinical practice. Despite these challenges, EV-based biomarkers have the potential to transform current strategies for AAA diagnosis, screening tests, and risk assessment.
In therapeutic applications, EVs may contribute to suppressing AAA development. However, current evidence remains limited, and several challenges must be addressed, including standardizing EV sources, dosing, and administration protocols. Moreover, the precise mechanisms underlying EV-mediated suppression of AAA remain incompletely elucidated. In addition, the large amounts of EVs potentially required for human administration, together with the associated medical costs, represent significant barriers to clinical translation. Nevertheless, no disease-modifying pharmacological therapy has been definitively established to halt AAA progression, and EVs may represent a promising avenue for future therapeutic development.
Although research on EVs in AAA remains limited compared with other fields, their clinical potential is considerable. Further studies are required to advance their translation into future clinical applications.
Reviewer 2 Report
Comments and Suggestions for Authors
Takahashi and colleagues present a review of the Role of Extracellular Vesicles in AAA. I have the following comments:
- Since this review focussen on extracellular vesicles, the introduction should not approach the epidemiological aspects. It should focus on the
- Please add a PRISMA statement
- Please add a section on translational research or applications
- Please add diagnostic advantages in comparison to ultrasound or CT
Author Response
Author's Reply to the Review Report (Reviewer 2)
Since this review focussen on extracellular vesicles, the introduction should not approach the epidemiological aspects.
We appreciate the reviewer’s comment on the Introduction's scope. While this review primarily focuses on extracellular vesicles, we believe that a brief overview of AAA pathophysiology is essential to provide the necessary biological context for understanding the roles of EVs in disease development. In particular, AAA pathogenesis involves multiple cell types and complex molecular mechanisms, many of which have not yet been fully investigated in the context of EV biology. We therefore consider this background necessary to identify current knowledge gaps and justify the need for further EV-related research in AAA. In addition, we retained only minimal epidemiological/clinical information to highlight the clinical burden and unmet needs in AAA, which motivates EV-based biomarker and therapeutic research, rather than to provide a comprehensive epidemiological review. To further emphasize this point, we have also added a statement in the Conclusion. Therefore, we considered it necessary to include the pathophysiological background for clarity and completeness, and the relevant sections were not removed. We have ensured that the Introduction remains focused on EV biology and its relevance to AAA.
We have added the following statement to the Conclusion section.
Line 484
Consistent with the scope of this review focusing on EV biology in AAA, studies investigating EVs in AAA remain limited, with most reports focusing primarily on EVs derived from inflammatory cells and VSMC. However, AAA pathophysiology involves multiple cell types and complex molecular mechanisms. These cell-derived EVs are needed for further research.
Please add a PRISMA statement
We appreciate the reviewer’s suggestion regarding the PRISMA statement. Since this manuscript is intended as a narrative review and not a systematic review or meta-analysis, we did not add a PRISMA statement. Instead, we have revised the manuscript to indicate its narrative review clearly.
Line 88-
This narrative review summarizes current studies on EVs in AAA pathophysiology, biomarkers, and therapeutic approaches. In addition, based on evidence from other related research, we discussed the prospects for EVs in AAA research.
Please add a section on translational research or applications
We appreciate the reviewer for the valuable comment regarding the translational aspects of this review. Because the number of translational studies specifically in the AAA field remains limited, we did not create a separate section. Instead, we discussed translational research from other related fields within the therapeutic section to provide relevant context.
In addition, we have expanded this section of the manuscript by adding text and appropriate references on EV dosing and routes of administration, thereby strengthening the review's translational perspective.
Line425
Although the therapeutic potential of EVs for AAA has been demonstrated, several challenges remain for EV-based drug development, including issues related to administration routes, dosing, and off-target effects [99-101]. Intravenous administration is currently considered the most common method. In mice, systemically administered EVs have a short blood half-life of approximately 2 minutes, whereas in primates the circulation time is approximately 60 minutes [99, 100]. Following intravenous administration, EVs are rapidly taken up by organs and are not readily degraded in the circulation. At low doses, EVs are preferentially accumulated in organs such as the liver and spleen. However, higher doses lead to broader distribution and accumulation in additional organs [101].
This dose-dependent biodistribution presents a major challenge for therapeutic applications, as effective delivery of EVs to the aorta may require the administration of large quantities of EVs in humans. Therefore, several studies have focused on enhancing the specificity and efficiency of EV uptake by target tissues.
Please add diagnostic advantages in comparison to ultrasound or CT
We thank the reviewer for this helpful suggestion. We have added a description of the diagnostic advantages compared with ultrasound and CT in the revised manuscript. In particular, blood-based assays can be implemented in a high-throughput manner and may improve accessibility and participation in population-level screening, whereas US/CT require dedicated imaging resources. We also clarified that EV-based biomarkers could complement imaging by supporting AAA screening and rupture risk assessment.
Line 323
Nevertheless, the development of blood-based EV biomarkers could enable a more convenient screening approach compared with US or CT. Because blood tests can be implemented in a high-throughput manner, EV-based assays may facilitate population-level screening and improve accessibility. In addition, EV-based biomarkers may offer higher diagnostic accuracy than conventional blood tests and hold potential clinical value for AAA screening and rupture risk assessment.
Reviewer 3 Report
Comments and Suggestions for Authors
Dear authors, I found your research very interesting.
This review summarizes current studies on extracellular vesicles (EVs) in the pathophysiology, biomarkers, and therapeutic approaches of abdominal aortic aneurysms (AAA).
Some observations:
The review describes the cellular and molecular mechanisms related to the development of AAA and the role of extracellular vesicles in this pathology.
1. In Figure 1, the authors are advised to improve the image resolution and increase the font size. The caption for Figure 2 briefly describes the image shown, while Figure 1 does not include a description; a brief description should be added.
2. In Figure 3, the authors are advised to increase the image resolution. Some of the figures shown are too small, and the size of the figures is not consistent. Increasing the font size is also recommended.
3. You state that “EVs have attracted considerable attention as therapeutic tools in AAA research.” Considering the clinical application of this therapy, what would be the limitations of using EVs? Would they be stable for a certain period of time? What would be a possible pharmaceutical form in which EVs could be administered?
I found your conclusion appropriate.
Author Response
Author's Reply to the Review Report (Reviewer 3)
This review summarizes current studies on extracellular vesicles (EVs) in the pathophysiology, biomarkers, and therapeutic approaches of abdominal aortic aneurysms (AAA).
Some observations:
The review describes the cellular and molecular mechanisms related to the development of AAA and the role of extracellular vesicles in this pathology.
- In Figure 1, the authors are advised to improve the image resolution and increase the font size. The caption for Figure 2 briefly describes the image shown, while Figure 1 does not include a description; a brief description should be added.
As suggested by the reviewer, we have improved the image resolution and increased the font size in Figure 1. In addition, a brief description has been added to the caption of Figure 1.
Figure 1 legend -
Figure 1. The normal aorta exhibits a three-layered structure with well-preserved elastic fibers. In contrast, abdominal aortic aneurysm (AAA) is characterized by extracellular matrix (ECM) degradation and infiltration of inflammatory cells, accompanied by the formation of an intraluminal thrombus (ILT). AAA development is driven by multiple pathological processes, including inflammatory cell infiltration, vascular smooth muscle cell (VSMC) apoptosis and phenotypic switching, and ECM degradation. These pathological changes involve complex interactions among diverse cell types and molecular mechanisms. The figure was prepared using BioRender (www.biorender.com). CXCL, C-X-C motif chemokine ligand; DAMPs, damage-associated molecular patterns; EC, endothelial cell; AoAF, aortic adventitial fibroblasts; HMGB-1, high mobility group box 1; IFN-γ, interferon-γ; IL, interleukin; MCP-1, monocyte chemoattractant protein-1; MMP, matrix metalloproteinases; NET, neutrophil extracellular trap; NLRP3, NOD-like receptor protein 3; ROS, reactive oxygen species; TIMP2, tissue inhibitor of metalloproteinase 2; TNF-α, tumor necrosis factor-α.
2. In Figure 3, the authors are advised to increase the image resolution. Some of the figures shown are too small, and the size of the figures is not consistent. Increasing the font size is also recommended.
As suggested by the reviewer, we have increased the image resolution of Figure 3. In addition, the figures have been resized for consistency, and the font size has been increased.
3.You state that “EVs have attracted considerable attention as therapeutic tools in AAA research.” Considering the clinical application of this therapy, what would be the limitations of using EVs? Would they be stable for a certain period of time? What would be a possible pharmaceutical form in which EVs could be administered?
We appreciate the reviewer for these important questions regarding the clinical application of EV-based therapies for AAA.
Regarding the limitations of EV-based treatments, we consider administration routes, dosing, and off-target effects to be significant challenges at this stage. We have added corresponding descriptions to the revised manuscript to clarify these limitations.
Regarding the stability of EVs, we have described the blood circulation half-life of systemically administered EVs and their cellular uptake. We also described whether EVs remain stable in circulation for a certain period and their uptake by target cells.
In addition, we have expanded the discussion to address the physicochemical stability of EVs during storage and handling prior to administration, including factors relevant to clinical translation.
Regarding the possible pharmaceutical form and route of administration, the manuscript previously described applications limited to endovascular methods. Given that intravenous administration is currently considered the most feasible and widely explored route for EV delivery, we have added text discussing it.
Line 425
Although the therapeutic potential of EVs for AAA has been demonstrated, several challenges remain for EV-based drug development, including issues related to administration routes, dosing, and off-target effects [99-101]. Intravenous administration is currently considered the most common method. In mice, systemically administered EVs have a short blood half-life of approximately 2 minutes, whereas in primates the circulation time is approximately 60 minutes [99, 100].
Following intravenous administration, EVs are rapidly taken up by organs and are not readily degraded in the circulation. At low doses, EVs are preferentially accumulated in organs such as the liver and spleen. However, higher doses lead to broader distribution and accumulation in additional organs [101].
This dose-dependent biodistribution presents a major challenge for therapeutic applications, as effective delivery of EVs to the aorta may require the administration of large quantities of EVs in humans. Therefore, several studies have focused on enhancing the specificity and efficiency of EV uptake by target tissues.
Line 458
Beyond biodistribution and targeting, the physicochemical stability of EV products during storage and handling (e.g., temperature control, freeze–thaw cycles, and potential aggregation or loss of bioactivity) is also an important consideration for clinical translation. Sterile aqueous suspensions are commonly used for intravenous injection/infusion in preclinical studies, whereas lyophilized formulations have been explored as a potential approach to improve shelf-life and facilitate handling [109]. While these formulation topics remain an active area of development, they represent practical constraints that must be addressed alongside dosing and targeting to enable EV-based therapeutics for AAA. In addition, scalable GMP-compatible manufacturing and robust quality control (e.g., identity, purity, potency, sterility, and batch-to-batch consistency) remain major hurdles for clinical translation of EV-based therapeutics [110].